# A Comprehensive Review on the Heavy Metal Toxicity and Sequestration in Plants

**DOI:** 10.3390/biom12010043

**Published:** 2021-12-28

**Authors:** Riyazuddin Riyazuddin, Nisha Nisha, Bushra Ejaz, M. Iqbal R. Khan, Manu Kumar, Pramod W. Ramteke, Ravi Gupta

**Affiliations:** 1Department of Plant Biology, Faculty of Science and Informatics, University of Szeged, Kozep fasor 52, H-6726 Szeged, Hungary; riyazkhan24992@gmail.com; 2Faculty of Science and Informatics, Doctoral School in Biology, University of Szeged, H-6720 Szeged, Hungary; 3Department of Integrated Plant Protection, Faculty of Horticultural Science, Plant Protection Institute, Szent István University, 2100 Godollo, Hungary; nisha3005n@gmail.com; 4Department of Botany, Jamia Hamdard, New Delhi 110062, India; bushra.libra91@gmail.com (B.E.); iqbal.khan@jamiahamdard.ac.in (M.I.R.K.); 5Department of Life Science, Dongguk University, Seoul 10326, Korea; manukumar007@gmail.com; 6Department of Life Sciences, Mandsaur University, Mandsaur 458001, India; pwramteke@gmail.com; 7College of General Education, Kookmin University, Seoul 02707, Korea

**Keywords:** heavy metals, ROS, redox status, photosynthesis, antioxidants, sequestration

## Abstract

Heavy metal (HM) toxicity has become a global concern in recent years and is imposing a severe threat to the environment and human health. In the case of plants, a higher concentration of HMs, above a threshold, adversely affects cellular metabolism because of the generation of reactive oxygen species (ROS) which target the key biological molecules. Moreover, some of the HMs such as mercury and arsenic, among others, can directly alter the protein/enzyme activities by targeting their –SH group to further impede the cellular metabolism. Particularly, inhibition of photosynthesis has been reported under HM toxicity because HMs trigger the degradation of chlorophyll molecules by enhancing the chlorophyllase activity and by replacing the central Mg ion in the porphyrin ring which affects overall plant growth and yield. Consequently, plants utilize various strategies to mitigate the negative impact of HM toxicity by limiting the uptake of these HMs and their sequestration into the vacuoles with the help of various molecules including proteins such as phytochelatins, metallothionein, compatible solutes, and secondary metabolites. In this comprehensive review, we provided insights towards a wider aspect of HM toxicity, ranging from their negative impact on plant growth to the mechanisms employed by the plants to alleviate the HM toxicity and presented the molecular mechanism of HMs toxicity and sequestration in plants.

## 1. Introduction

Feeding an exponentially growing population in the era of global climate change is a serious challenge. Among different environmental factors, abiotic stressors are the major factors affecting crop yield and the livelihood of mankind [1]. In particular, heavy metal (HM) toxicity has emerged as one of the most serious threats to the crop production that might become even more prevalent in the coming decades. HMs are a group of 52 metals including lead (Pb), manganese (Mn), copper (Cu), nickel (Ni), cobalt (Co), cadmium (Cd), mercury (Hg), and arsenic (As) which directly affect the plant performance in a concentration-dependent manner [2,3]. When present in adequate amounts, both micro and macronutrients maintain the functionality of key enzymes and regulate the metabolic pathways including photosynthesis, DNA synthesis, protein modifications, sugar metabolism, and redox homeostasis. However, at higher concentrations, these exhibit toxicities and can be lethal for all the lifeforms including plants as well as animals.

Increased anthropogenic activities such as mining, modern agriculture practicing, fertilizer application, extensive use of groundwater for irrigation, sewage disposal, and industrialization have disturbed the distribution of these HMs, leading to their accumulation at specific sites [4,5]. Accumulation of HMs consequently affects soil health and texture because of the alteration in soil pH and the ratio of essential/or nonessential elements, which is a global concern for agricultural production [6,7]. Moreover, absorption of these HMs from the contaminated soil and their accumulation, especially in food crops, are a serious concern for human health. It has been reported that these HMs, following their absorption from the contaminated soil, can accumulate in the edible plant parts and thus enter the food chain with a potential health risk to humans and animals [8,9].

Plants growing in HM-contaminated soils show visible symptoms like stunted growth, chlorosis, root browning, and sometimes even death [10,11]. Moreover, HM toxicity hinders the metabolic, cellular, and genetic potential of the plants required for normal growth and development. However, some of the plants can cope with a moderate concentration of HM(s) by reducing their uptake, their compartmentalization into vacuoles, their sequestration through phytochelatins (PC)/metallothioneins (MT), and activation of antioxidant defense [12]. Since the accumulation of reactive oxygen species (ROS) is one of the most common effects of stress conditions in plants, activation of antioxidant defense mechanism is a major strategy to overcome the adverse effects of stress conditions. Nevertheless, the effectiveness and initiation of a particular defense response depend on the plant and type of HMs as well as their concentration and duration of exposure. Therefore, the need of the hour is to understand the effect of HM toxicity on crop plants in order to find a scientific way for improving crop yield in HMs contaminated soil, which is going to be more prevalent in near future. The present review was an upfront effort to integrate the recent understanding of morphological, physiological, and biochemical mechanisms of HMs-induced plant stress responses in order to present the molecular mechanism of HM toxicity and tolerance in plants.

## 2. HM Toxicity Induces Morphological, Anatomical, and Physiological Changes in Plants

Plants growing in HM contaminated soils show visible symptoms as the uptake and accumulation of HMs in plant tissues alter the morphology and overall health of the plants (Figure 1).

The effect of HM toxicity can be observed on almost all the plant tissues and at all the stages of plants’ life cycle starting from seed germination to senescence; however, these effects are more pronounced during seed germination and root growth. Since seed germination is sensitive to physicochemical conditions of the rhizosphere, decline in seed germination and vigor and subsequent seedling growth have been reported under HM toxicity in most cases [13]. In particular, Pb-toxicity has been shown to induce adverse effects on seed morphology, physiology, germination, and early crop growth in a variety of crops including *Raphanus sativus* [14], *Lens culinaris* [15], *Oryza sativa*, *Hordeum vulgare* [14], *Elsholtzia argyi* [16], *Spartina alterniflora* [17], *Vigna radiata* [18], *Medicago sativa* [19], and *Zea mays* [20], as reviewed previously [21]. In addition to Pb, exogenous application of other HMs also exhibits similar effects on seed morphology, physiology, and germination. For instance, a combined treatment of Cu and Cd to *Solanum melongena* severely reduced seed germination, seedling growth, and number of lateral roots [22]. Although the exact molecular mechanism of this HM toxicity-induced changes in seed physiology is not well understood, a growing body of evidence suggests that HMs cause inhibition of various enzymes which results in these changes. For instance, Hg-mediated inhibition of seed germination and embryo growth was found to be because of the direct interaction of Hg with the –SH group(s) of the proteins, resulting in the formation of a S-Hg-S bridge causing uneven modification in protein structures and thus loss of enzymatic activities [23]. Since the functions of certain enzymes such as amylases and proteases are crucial for seed germination, their inhibition affects the process of seed germination and subsequent hypocotyl and radical growth [16,17].

Apart from the seeds, changes in the root architecture have also been observed in the plants growing in the HM contaminated soils. In particular, decreased root elongation and increased lateral root formation have been reported in the presence of various HMs such as Cu, Pb, Cr, Zn, and Cd stress in several plants including *Arabidopsis thaliana* [24], *Triticum aestivum* [25,26], *Sesbania rostrata, Sesbania cannabina* [27], *Pinus sylvestris* [28], and *Lupinus luteus* [29], among others. In *Vigna unguiculata*, localized swelling was reported behind the root tips because of the initiation of the lateral roots and bending of some root tips along with the reduction in fresh biomass and decreased root growth under Pb-toxicity [30]. It has been suggested that HMs, particularly Zn, accumulate in the endodermis and pericycle and trigger the formation of lateral roots [31]. This formation of the lateral roots under HM toxicity is the initial symptom of HM toxicity, which subsequently impairs absorption and conduction of water which, in turn, results in lower transport of photosynthates to roots. Since roots receive lesser amounts of photosynthates during HM toxicity, the growth of primary and secondary roots are inhibited, resulting in a lower specific root length and a high root/shoot area ratio [32].

Subsequent research showed that these HM toxicity-induced changes in root architecture are partially remodeled by the action of indole acetic acid (IAA), and phenols produced in the presence of HMs [26,33,34]. Genes involved in IAA biosynthesis were reported to be upregulated, leading to the increased content of auxin and auxin/cytokinin ratio under HM toxicity, which acted as key factors responsible for determining morphological changes in roots [35]. Moreover, emerging evidence also highlights the role of Nitric oxide (NO) in inhibition of root meristem growth as observed under Cd stress in Arabidopsis [36]. Cd-induced NO accumulation is involved in the inhibition of auxin transport to the root apex, leading to a reduction in the meristem size. Moreover, an auxin efflux carrier protein PINFORMED1 (PIN1), responsible for root meristem growth and elongation, was found to be downregulated under HM toxicity, further indicating the fine-tuning of auxin transport under HM toxicity [37]. These observations were further supported by the results of [38], showing a similar NO-auxin cumulative root meristem inhibitory effect under Cu stress.

Along with the root inhibition, stunted growth of the plants has also been observed under HM toxicity [39]. Transport of HMs from contaminated soils via roots to aerial parts and their accumulation in plants cells interfere directly with the cellular metabolism of shoot causing a reduction in the height as observed in a variety of crop and non-crop plants including mung bean [40,41], oats [42], *Curcumas sativus*, *Lactuca sativa*, *Panicum miliaceum* [43], wheat [44], *Jatropha curcas* [45], maize [46,47,48], rice [49], and poplar [50]. Although the exact molecular mechanism of HM-induced growth inhibition is currently unknown, it has been reported that some of the HMs such as As and Hg induce loss of cell turgidity, a decline in mitotic activity [51], and inhibition of cell elongation as a result of their selective binding to the –SH group of proteins involved in the regulation of cell division and replacement of phosphate group of ATPs, and thus are involved in the reduction in plant growth under HM toxicity [52]. Besides, As has also been shown to dismantle the chlorophyll molecules by interacting with the central atom (Mg) of the porphyrin ring, leading to the inhibition of photosynthesis which affects the overall growth as reported in the sunflower seedlings [53,54]. Moreover, Cu toxicity mediated enhanced lignification in both shoots and roots has also been reported, which subsequently resulted in reduced biomass accumulation due to impairment in the cell development, leading to a greater formation of trichomes on the leaves and stems probably to sequester excess Cu [55,56].

In the case of leaves, HM toxicity affects the area, number, pigmentation, and thickness of leaves by impairing plant-water relations which, in turn, affects various physiological processes such as transpiration and photosynthesis [30,57]. In general, a drop in leaf number, area, and biomass has also been reported in a number of plants including *Albizia lebbeck* [44,58], *Arabidopsis thaliana* [26], *Spinacea oleracea* [59], *Brassica oleracea* [60], *Oryza sativa*, *Acacia holosericea*, *Leucaena leucocephala* [41,61], *Prosopis laevigata* [62] *Arabidopsis thaliana* [26], and *Spinacea oleracea* [59] under HM toxicity. In the case of sugarcane, a lower concentration of Cr (40 ppm) induced leaf chlorosis while a higher concentration (80 ppm) caused necrosis [63]. Moreover, browning of leaf tip of *Phaseolus vulgaris* [64] due to Hg toxicity and small brittle purple leaves of *Jatropha curcas* [16] in the presence of Pb have also been reported under HM-toxicity. Under Zn and Cd- Cu stress, a decrease in the stomatal index was observed in *Beta vulgaris* [65] and *Sorghum vulgaris* [66]. In some plants such as *Helianthus annuus* and *Vigna radiata*, an increased number of stomata has been reported in the presence of Pb and As at the earlier stages of HM toxicity, which was accompanied by the development of arrested, fused, and abnormal stomata in *Vigna radiata* [67,68]. Similarly, Zn and Cd toxicity resulted in the development of anomalous and non-functional stomata in *Zea mays* [69]. Further, HM toxicity also results in the reduction of parenchymatous tissues and xylem vessels along with the development of smaller mesophyll tissue as observed in *Brachiaria decumbens* under Cu stress [70]. All these HM-toxicity-induced changes in the stomata, xylem vessels, parenchymatous and mesophyll cells subsequently alter the plant-water relation and thus are, at least partially, responsible for the decreased leaf growth.

## 3. HM Toxicity Negatively Influence the Photosynthesis

Since the accumulation of HMs in the plants leads to adverse changes in the primary photosynthetic organ leaves, a decline in the rate of photosynthesis can be presumed. Literature hints that the impact of HMs on photosynthetic machinery depends on the reactivity and concentration of HMs in leaves, which subsequently affects light capture, electron transport, and activities of photosynthetic enzymes including RuBisCO (Figure 2) [71,72,73].

For instance, a reduction in maximal photochemical efficiency (Fv/Fm), effective quantum yield of photosystem II (φPSII), and photosynthetic electron transport rate (ETR) values were reported during Cd stress in lettuce plants, indicating inhibition of photosynthesis [74], while in wheat, Cd toxicity was manifested by the inhibition of the oxygen evolution, the decline in PSI activity, and chlorophyll fluorescence [75]. Likewise, Mazur et al. (2016) reported that Tl toxicity resulted in a decrease in the plant and leaf size, the number of discolored and necrotic leaves, changes in the mesophyll structure, and a 50% decline in the PSI and PSII activities in white mustard (*Sinapis alba* L.) [76]. A similar result was also reported under Cr toxicity, where a reduced number of active reactions centers of photosystem II together with the loss of starch grains and appearance of plastoglobuli in the chloroplast of *Spirodela polyrhiza* were reported [77]. Other plants experiencing toxicity of a variety of HMs also showed a similar drop in the photosynthesis and all of these studies collectively suggested that the decline in photosynthesis is majorly because of the decrease in the chlorophyll concentrations. Based on the literature evidence, it can be concluded that HMs can target chlorophyll in any of three ways: (1) by enhancing the activity of chlorophyllase enzyme [73]; (2) by causing oxidation/reduction of chlorophyll molecules by the HM toxicity induced ROS [78,79,80]; and (3) inhibition of chlorophyll biosynthesis as some of the HMs such as Hg, Cu, Pb, Ni, Cd, and Zn are able to replace the Mg in the porphyrin ring of the chlorophyll molecules [45,74,81,82,83,84,85]. Besides, phaeophytization of chlorophyll molecules because of the replacement of Mg^2+^ ions by H^+^ ions has also been reported in *Eudorina unicocca* and *Chlorella kessleri* under Cr-toxicity [86].

In addition to targeting the chlorophyll molecules, changes in chloroplast and thylakoid membranes have also been reported in presence of several HMs because of the HM-stress-induced ROS production (Figure 2). For instance, swelling of thylakoids, degradation of internal chloroplast membranes, and loss of chloroplast membrane integrity were observed during Pb/Cd toxicity in barley [87] and *Lemna minor* [88]. Likewise, the poor lamellar arrangements with few grana with widely placed thylakoids were reported in *Limnanthemum cristatum* under Cr toxicity [89]. In addition to causing the peroxidation of chloroplast membranes, HM-stress-induced ROS production also minimizes the uptake of crucial elements required for the synthesis of photosynthetic pigments such as Mg, K, Ca, and Fe [90,91] as reported under various HM stresses such as in Pb, Cd, and Cr in *Brassica napus* [92,93], Cu, Pb and Cd stress in wheat [94,95], and Al, Pb, Cr, Cd and Zn toxicity in sorghum [96,97]. Moreover, the inactivation of photosynthetic enzymes such as RuBisCO and other Calvin cycle enzymes has also been reported under HM toxicity (Figure 2). In barley, it was shown that Pb/Cd-induced ROS interacted with the thiol group of enzymes and ultimately hindered the functions of chloroplast, including photosynthesis [87]. A similar observation was also reported during Cd toxicity where Cd altered the protein structure via ROS induction and inhibited enzyme activity by interacting with the sulfhydryl and carbonyl groups of the proteins and replacing essential co-factors of enzymes [98,99].

Taken together, it can be concluded that HM toxicity reduces the net photosynthesis rate by inhibiting both light and dark reactions of photosynthesis. It is quite evident that inhibition of light reactions during HM toxicity is because of the decreased efficiencies of PS I and PSII, which is a combined effect of reduced chlorophyll concentration and destabilized chloroplast and thylakoid membranes. On the other hand, inhibition of dark reactions is majorly because of the decreased activities of enzyme associated with the Calvin cycle [72,100,101].

## 4. Antioxidant Enzymes Alleviate the HM Toxicity Induced Oxidative Stress

HM toxicity induces the production of excessive ROS, which interact with macromolecules such as DNA, proteins, and lipids, leading to a series of vicious processes together termed as “oxidative stress” [102,103,104]. A substantial number of investigations suggests that HMs alter cellular redox equilibrium in the five conceivable succeeding ways: (1) by shifting redox potential to more oxidized values [105]; (2) by directly producing ROS through Fenton-like reactions and the Haber-Weiss cycle [103,106]; (3) by consuming GSH, which is involved in the synthesis of PCs required for the chelation and sequestration of HMs into vacuoles via ABC transporters [107]; (4) by directly inhibiting the activities antioxidant enzymes by exchanging essential cations from the specific binding sites and targeting their -SH groups [108]; and (5) by inducing the activity of NADPH oxidases [108].

Redox homeostasis is largely dependent on the balance of reduced glutathione (GSH)-oxidized glutathione (GSSG), which is majorly maintained by the activity glutathione reductase (GR) [109]. Under normal conditions, glutathione and ascorbate are majorly available in the reduced form and are involved in the detoxification of oxidants at different cellular compartments including cytoplasm, endoplasmic reticulum, vacuoles, mitochondria, chloroplast, and peroxisome [110,111]. At first, it was reported that Al toxicity elevated the level of GSH as an earlier response and provided HM-stress tolerance in *Pisum sativum*, which indicated the changes in cellular redox levels in response to HM toxicity and subsequent accumulation of antioxidants to maintain the redox homeostasis [112]. Further, transcript levels of GR were found to be elevated in Cr treated *Brassica juncea* and maize seedlings [113,114]. However, this increased *GR* expression with high GSH/GSSG was not sufficient to protect the cell from Cr-induced oxidative damage in maize [114]. Yet, increased GR activity has been reported under moderate HM-stress conditions in a variety of plants including pea [115], wheat [116], alfalfa [117,118], *S. vulgaris* [119], and Arabidopsis [120]. In addition to the GR, a substantial number of investigations suggest that GRXs also play a key role in HM toxicity tolerance in plants (Figure 2). GRXs are GSH-dependent disulfide oxidoreductases that catalyze the reduction of disulfide bonds of their substrate proteins and participate in oxidative stress responses in plants [121,122]. It was reported that the ectopic expression of chickpea (*Cicer arietinum* L.) glutaredoxin (*CaGrx*) gene in *A. thaliana* elevated GSH levels and maintained the redox homeostasis under the toxicities of various HMs including AsIII, AsV, Cr(VI), and Cd [123]. Likewise, *OsGrx* genes were also found to be involved in detoxification of As and provided HM stress tolerance by regulating/maintaining GSH level and/or GSH recycling. Transgenic *A. thaliana* plants overexpressing *OsGrx_C7* and *OsGrx_C2.1* genes showed elevated Grx activity and a higher intracellular GSH content under As stress as compared with wild-type (WT) [124]. These results altogether indicate that GRXs are able to maintain ROS homeostasis by increasing the production and recycling of GSH.

In addition to GR and GRX, functions of other enzymatic antioxidants in HM-stress tolerance have also been highlighted. For instance, Chowardhara et al. (2020) reported an accumulation of non-enzymatic antioxidants such as proline, ascorbate, and glutathione and increased activities of enzymatic antioxidants including SOD, CAT, GST, GR, APX, and POX under Cd toxicity in the three cultivars of *B. juncea*, indicating a pivotal role of these antioxidants in the HM-stress tolerance [125]. Further, overexpression of *Triticum aestivum* catalase (*TaCAT3)* gene provided As stress tolerance via enhancing the catalase activity [126]. Interestingly, overexpression of *TaCAT3-B* genes in *E. coli* also resulted in higher tolerance towards As^III^ and As^V^ toxicity in the transgenic lines [126]. Moreover, while comparing different flax (*Linum usitatissimum* L.) genotypes for their Pb-stress tolerance, Pan and co-workers observed that the Milas genotype, showing increased expression of various genes encoding for antioxidants such as *LuSOD1*, *LuPOD1*, and *LuPOD2*, was more tolerant to Pb-toxicity than others [127].

## 5. Tolerance to HM Toxicity Is Mediated by a Complex Signaling Network

Plants under HM toxicity exhibit biochemical, physiological, and molecular changes as a part of their tolerance response, and understanding these responses is crucial to induce HM tolerance in sensitive crop plants (Figure 3).

A growing body of evidence suggests a crucial role of various signaling molecules including H_2_S and Cys in HM stress tolerance. Both H_2_S and Cys are essential sulfur metabolism products that have emerged as novel gasotransmitters and redox signal molecules. At first, an upregulation of H_2_S biosynthesis genes such as L-Cysteine desulfhydrase (*LCD)* and L-Cysteine desulfhydrase 1 *(DES1)* was observed under Cd toxicity, suggesting a release of endogenous H_2_S under HM-toxicity. Therefore, to confirm the roles of these two signaling molecules, mutants of H_2_S (*lcddes1-1)* and Cys (*oasa1*, O-acetylserine(thiol)lyase isoform A1) biosynthesis genes were developed in Arabidopsis which were defective in the production of H_2_S and Cys. Interestingly, these mutants showed decreased Cd stress tolerance as compared with the WT, which confirmed the pivotal roles of these two molecules in HM-stress tolerance [128]. Similarly, Arabidopsis *oasa1-1* and *oasa1-2* mutants, which are defective in Cys production, showed significantly reduced Cd tolerance [129]. Altogether, it has been proposed that H_2_S and Cys participate in suppressing HM stress in plants via inhibiting ROS burst by inducing the expression level of MT genes, alternative respiration capacity, antioxidant activity, and GSH accumulation following the enhancement of the level of PCs genes (Figure 2) [128]. In addition to PCs, several other genes have been shown to play a positive role in mitigating HM toxicity (Table 1).

For instance, upregulation of acyl-CoA-binding proteins *ACBP1* and *ACBP4* was reported under Pb^2+^ stress and the overexpression of *ACBP1* gene improved tolerance to Pb^2+^ in Arabidopsis [130,131]. Moreover, transgenic tobacco overexpressing a truncated *NtCBP4* was more tolerant to Pb^2+^ as compared with plants overexpressing the full-length *NtCBP4* gene [132]. Moreover, the disruption of the homologous Arabidopsis *CNGC1* gene (cyclic nucleotide-gated ion channel) conferred tolerance to Pb toxicity. *NtCBP4* and *AtCNGC1* are both components of a transport pathway that facilitates the entry of Pb^2+^ into plant cells and thus disruption of these genes could be an important strategy for inducing the HM-stress tolerance in plants by limiting the uptake of these HMs inside the plant cells.

Transcription factors such as GRAS (GRAS domain family), myeloblastosis protein (*MYB*), *C2H2*, bHLH, *NAC*, basic leucine zipper (bZIP), ethylene-responsive (ERF), and *WRKY,* among others, are emerging out to be the regulators of HM-stress tolerance in plants [133,134,135,136,137]. For instance, transcriptome analysis of kenaf (*Hibiscus cannabinus* L.) showed enhanced expression of various transcription factors, including *WRKY*, *GRAS*, *MYB*, *bHLH*, *ZFP*, *ERF*, and *NAC* under Cd stress [137]. Similarly, activation of a large number of WRKY genes along with C2H2 zinc finger protein, AP2 domain-containing protein, MYB, and HSF have been reported in Arabidopsis under Pb and Cd toxicities [138,139] and in rice under Cd-stress conditions [133,134]. In Arabidopsis, several transcription factors including a Zinc-Finger protein ZAT6 were found to be induced during Cd stress and provided tolerance via the glutathione-dependent PC pathway [140].

## 6. Sequestration and Compartmentalization: Plants’ Way to Alleviate the HM Toxicity

During their growth and development, plants absorb essential elements such as carbon (C), nitrogen (N), potassium (K), zinc (Zn) as well as non-essential elements such as Cd, Hg, and Pb, among several others [141]. Although trace quantities of non-essential elements are beneficial for plant growth, excessive accumulation of these elements adversely affects various physiological and metabolic processes and deteriorates plant growth and development [141]. Metal accumulation in plants is majorly governed by the two processes including their uptake inside the plant cells and their translocation from roots to other parts [142]. In most of the cases, the major percentage of the HMs in plants is transported from root to stem and thus the concentration of HMs in the aboveground parts is higher than the roots; however, the most toxic HMs are commonly not transported in non-hyperaccumulating plants and the highest concentration is usually found in the roots [79]. For instance, in tomato plants, the highest concentrations for Cu, Ni, Cr, Mn, and Pb were reported in order of root > leaf > stem > fruit [143].

HM stress-tolerant and hyperaccumulator plants get rid of the unused and extra amount of metal ions by effluxing and/or compartmentalization majorly in the vacuole with the help of two vacuolar proton pumps, including an ATPase and a Ppas (Figure 2) [144,145,146]. For instance, sequestration of Zn in the hyperaccumulator plants has been majorly observed in the vacuoles of epidermal cells and trichomes of mesophyll cells, as shown in *Thlaspi caerulescens* [147] and in *Arabidopsis helleri* respectively [148,149]. In addition, Zn can also be accumulated, although to a lesser extent, in the cell wall and cytosol in leaves of another hyperaccumulator plant *P. griffithii* [150]. Apart from the Zn, Ni was also found to be accumulated in the vacuoles of a Ni hyperaccumulator plant *Alyssum serpytllifolium* [151]. Similar findings were also reported in the case of tolerant plants such as *B. juncea, Silene vulgaris* [152,153,154], and *Brassica napus* [149,155] where an accumulation of Cd has been reported majorly in the epidermal and mesophyll cells.

Apart from sequestration, plants minimize the HM toxicity by limiting the absorption of HMs from the soil through secretion of chelating compounds in the root hairs [156]. Plants synthesize cystine-rich metal-binding peptides known as PCs and MTs to chelate the HMs. Among different HMs, Cd has been identified as the most potent inducer of PC synthesis in plants [79,117,157,158]. These PCs bind to the HMs and sequester them into the vacuoles; however, PCs accumulation does not always have a beneficial effect on plants. For instance, rice plants expressing *TaPCS1* from *Triticum aestivum* were found to be Cd sensitive and showed enhanced Cd accumulation in shoots [159]. Similar to the PCs, MTs also act as biochelators and can directly bind to various HMs such as Zn, Cu, Cd, and Ni, among others. These MTs are localized in the membrane of the Golgi apparatus and growing evidence also indicates their role in maintenance of ROS homeostasis during HM toxicity. For instance, ectopic expression of different MT genes including type 1, type 2, and type 3 genes from a variety of plants such as rice, *B. juncea*, and *Elsholtzia haichowensis* has been shown to enhance the tolerance to Cu and/or Cd stress in transgenic plants [160,161,162]. This MT-induced enhanced HM-toxicity tolerance was because of the increased SOD [163] and POD [164] activities and decreased production of hydrogen peroxide (H_2_O_2_) that protected the transgenic plants from the HM-toxicity-induced oxidative stress. Similar to the MTs, some of the beneficial elements such as silicon (Si) and selenium (Se) also participate in the HM-stress tolerance by limiting the uptake of HMs and enhancing the activities of antioxidant enzymes [165]. There are now ample reports which have shown that the exogenous application of either of these two metals enhances plants’ tolerance to HM toxicity [166]. For instance, exogenous treatment of Si has been shown to decrease the uptake, transport, and accumulation of Cd in various plants such as peanut, *Cucumis sativus*, cotton, and *Brassica chinensis* and mitigate the toxic effects of Cd toxicity by reducing the electrolyte leakage, MDA and H_2_O_2_ contents and improving the activities of antioxidant enzymes such as CAT, SOD, and POD [167,168,169,170].

Apart from these proteins and metals, various secondary metabolites also act as metal chelators and are involved in the sequestration of HMs to vacuole to provide HM-stress tolerance (Figure 2). Su et al. (2020) investigated the role of lignin in HM-stress tolerance and found that two lignin-defective mutants (*ldm1* and *ldm2*) of rice showed dwarf shoots and roots due to higher levels of Cu accumulation in the roots and leaf sheaths tissues, suggesting that the promotion of lignin synthesis in plants may be an adaptive strategy for Cu adaptation [171]. Interestingly, a higher accumulation of phenolics and amino acids was found to be associated with Co and Cu stress tolerance. It was observed that exposure of Co and/or Cu significantly reduced chlorophyll content and photosynthetic and transpiration rates of the two barley genotypes. In the comparison of a single treatment of Co or Cu, their co-application significantly attributed the higher accumulation of phenolics, including cinnamic acid and benzoic acid derivatives, together with free amino acids in tolerant genotypes (Yan66) than in susceptible genotype (Ea52). Moreover, phenolic compounds also participated in the vacuolar sequestration of Cd which enhanced Cd tolerance in *Thlaspi caerulescens* [172]. In *Jatropha curcas*, phenylalanine ammonia-lyase (*JcPAL*) genes were identified in the metal detoxification mechanism along with the metallothioneins (*JcMT2a*). Interestingly, the transcript levels of *JcPAL-R* and *JcPAL-L* were upregulated with increasing Pb dose in a dose-dependent manner. Overall, their results suggested that along with the steadiness of growth traits, upregulation of metal transporter, and antioxidant defense system and higher accumulation of antioxidants including flavonoids and phenolics also participate in alleviating the HM-toxicity [173]. Similarly, the phenylalanine ammonium lyase-related gene (*HvPAL*) was found to be highly upregulated along with the genes (*HvMDH* and *HvCSY*), involved in the biosynthesis of malate and citrate, under the combined treatment of Co and Cu in barley. Some organic acids such as citrate and malate are known to participate in HM tolerance directly or indirectly [174]. The increase of these organic acids might be responsible for energy production under the combined treatment to fulfill the higher energy demand by metal transporters to improve Co and Cu tolerance [175]. These metal transporters, including ABC type transporters, are a group of organic solute transporters that help in vacuolar sequestration of “PC-metal complexes” and play an important role in metal ion homeostasis and tolerance [176,177]. For instance, it was shown that “PC-metal complexes” of metal ions such as Zn^2+^, Cu^2+^, and Mn^2+^ can be transported into the vacuole through the putative ABC transporter(s) [178]. A recent study has highlighted that H_2_S functions by regulating the expression levels of genes of the ATP-binding cassette (ABC) superfamily to provide HM-stress tolerance [179,180]. In rice, H_2_S was found to alleviate the Al^3+^ toxicity by enhancing the expression levels of ABC transporters, including *OsSTAR1* and *OsSTAR2* genes which prevent entering of Al into the cytoplasm [181]. Besides, some other studies have also highlighted the pivotal roles of ABC and other transporters in HM-stress tolerance. For instance, upregulation of wheat ABCC transporters including *TaABCC3*, *TaABCC4*, *TaABCC11*, and *TaABCC14* was observed under Cd toxicity [182]. In *Brassica napus*, ABCC transporters *BnaABCC3* and *BnaABCC4* were upregulated under Cd stress and enhanced Cd tolerance by limiting the entry of Cd inside the cells and their phytochelatin-mediated detoxification [183]. Similar observations have been observed in Arabidopsis where ABCC transporters *AtABCC3* and *AtABCC6* were found to be involved in phytochelatin-mediated Cd tolerance during seedling development [184,185]. In addition to these expression studies, overexpression studies have further supported the role of ABC transporters in HM-stress tolerance. For instance, overexpression of *FvABCC11* gene increased Cd tolerance in strawberry [186] and *AtABCC1* and *AtABCC2* conferred tolerance to Cd and Hg in Arabidopsis through the vacuolar sequestration of these HMs [187]. It was determined that *OsABCC1* regulated As-stress tolerance by sequestering As in vacuoles of the phloem companion cells of the nodes in rice grains [178]. Similarly, a heavy metal transporter gene *AtPDR8* (an ABC transporter) was identified as a cadmium-extrusion pump in Arabidopsis to mediate resistance against Cd and Pb toxicity [188].

Along with the ABC transporters, involvement of other transporters including multidrug and toxic compound extrusion (MATE) family proteins [189,190], zinc-iron permease (*ZIP)*, iron-regulated transporters (*IRT1*) [190], natural resistance-associated macrophage proteins (NRAMP), copper transporters (Ctr/COPT), and heavy metal ATPase (HMA) transporters [191] has also been highlighted in tolerance to HM toxicity. These metal transporters are involved in mediating metal intake and translocation in plants [192]. For instance, IRT1 (a Fe transporter gene) that belongs to the ZIP family has been identified in Arabidopsis that leads to high-affinity Fe uptake under Fe-deficient conditions [193] along with uptake of other heavy metals such as Cd and Zn [194]. Other ZIP metal transporters such as *ZIP1* and *ZIP2* were identified as Zn and Mn transporters in roots that mediate remobilization of Zn and/or Mn from the vacuole to root stele [195].

Metal transporters such as cation diffusion facilitator (CDF) transporters, ferroportins, VIT1/CCC1 also play a crucial role in the detoxification of HMs in plants and prominently remove transition metals out of the cytoplasm in the plant cell [196]. ZAT transporters (a CDF transporter) in Arabidopsis such as *ZAT1* showed a significant role in Zn metal sequestration as transgenic lines with overexpression of *ZAT1* exhibited increased Zn resistance [197]. It was hypothesized that ZAT is involved in vesicular or vacuolar sequestration of HMs such as Zn and thus is involved in enhancing Zn tolerance in plants [197]. The expression of *ZTP1* (a ZAT gene) in the Zn hyperaccumulators such as *Thlaspi caerulescens* in response to calamine (source of Zn, Pb, and Cd) and serpentine (rich in Ni) suggests the alleviatory role of *ZTP1* in plants during HM toxicity [197,198]. *ShMTP1*, a CDF transporter, confers Mn tolerance in Arabidopsis by sequestering metal to internal organelles and may function as an antiporter of Mn to enhance tolerance [199]. Several P_1B_-ATPases in Arabidopsis have been identified that majorly alleviate metal accumulation. For instance, ectopic overexpression of *AtHMA4* in Arabidopsis mediated plant growth and development by enhancing the root growth and root-to-shoot metal translocation in the presence of toxic heavy metals (Zn, Co, and Cd) [200]. The expression of P-type ATPase transporters enhanced under Pb stress in *Brassica juncea* plants [201] and tomato seedlings [202]. The metal transporters including ABC, ZIP, NRAMP, Ctr/COPT, and HMA maintain metal homeostasis via activating different signaling components such as MAPK signaling and hormone and calcium signaling [203]. Metal tolerance proteins (MTP) are also considered to be the potential efflux transporters that extrude mainly Zn out of cytoplasm along with other metals (Mn, Fe, Cd, Co, and Ni) [204]. *MTP1* confers to Zn hypertolerance in plants during high Zn concentration [205]. Identification of undiscovered transporter genes, their functional characterization, and their manipulation can further aid in the production of hyperaccumulators as well as plants with enhanced phytoremedial potential. The HM-stress detoxifying transporters can be employed in growing applications of molecular-genetic technologies for a better understanding of heavy metal accumulation and tolerance in plants. These transporters either limit the entry of HMs into the cells or are involved in the vacuole sequestration of these HMs to improve HM-stress tolerance.

## 7. Plants Retaliate against HM Toxicity by Elevating the Levels of Compatible Solutes

HM toxicity obstructs the plant metabolic, cellular, and genetic potential required for normal growth and development. Cellular toxicity associated with the overproduction of reactive oxygen species (ROS) leads to induction of oxidative stress [206,207]. At the metabolic level, HMs interfere with various proteins, causing their inactivation, resulting in the repression of key physiological processes such as photosynthesis and respiration [104,208]. Metal toxicity provokes oxidative stress in plants which eventually leads to acceleration of the antioxidant system and osmoregulation via accumulation of osmolytes or compatible solutes which maintains the osmoticum of the cells [208,209,210]. In response to stress, plants produce various types of compatible solutes including proline, polyols, soluble sugars, and quaternary ammonium compounds (QACs) such as glycine betaine, proline betaine, alanine betaine, and polyamines, which participate in osmotic adjustment, stabilization of macromolecules, metal chelation, and ROS detoxification [211,212]. In addition, a direct role of some of these compatible solutes, including polyamines, organic acids, and amino acids, in the HM chelation has also been reported. For instance, exogenous application of polyamine spermidine (Spd) induced the activities of SOD and GPX and thus maintained the lower levels of superoxide anion (O_2_^−^) and MDA in leaves of *Malus hupehensis* under Cd toxicity [213]. Similar observations were also reported in rice where a protective role of Spd and spermine (Spm) were reported under Cd stress. It was reported that exogenous treatment of these polyamines reduced the CdCl_2_-induced oxidative stress in plants by lowering the MDA and H_2_O_2_ level due to augmented levels of SOD, GR, and APX [214]. In wheat, priming of seed with polyamines protected the seedlings from Pb stress and improved the growth and yield traits by improving the membrane stability index (MSI), relative water content (RWC), contents of photosynthetic pigments, osmoprotectants, nutrients, ascorbic acid, total glutathione, and activities of SOD and CAT [215,216]. Spd also improved the length of roots, shoots, and fresh weight of Cr-stressed seedlings of *Raphanus sativus* by increasing GSH, ascorbic acid, proline, glycine betaine total phenol, and antioxidant enzyme activities such as guaiacol peroxidase, CAT, SOD, and GR [217]. In addition, polyamines also function as signaling molecules and regulate plant stress responses under HM toxicity through ion homeostasis and transportation [218]. It was shown that pre-treatment of Spm reversed the Cu and Cd-induced ROS production and subsequent lipid peroxidation [219]. Similarly, pre-treatment or exogenous applications of Spm and Spd were found to be effective in alleviating the toxic effects of HMs as observed in a variety of plants including *Nymphoides peltatum* [220], *Typha latifolia* [221], *Raphanus sativus* [217,222], *Boehmeria nivea* [223], *Triticum aestivum* [215], *Vigna radiata* [224], *Salix matsudana* [225], *Vigna angularis* [226], and *Zea mays* [227].

Similar to the polyamines, organic acids such as tartrate, citrate, oxalate, malonate, aconitate, and malate have also been shown to bind with the ions of HMs via the chelation with carboxyl groups that act as oxygen donors in metal ligands events [228]. However, many studies revealed the involvement of OAs in the regulation of transport and internal chelation of HMs in plants. In the root tip of wheat plants, membrane-localized malate transport channels (ALMT) were found to be activated in response to a high concentration of Al^3+^, which resulted in the excretion of malate to limit the entry of Al^3+^ into roots via their chelation [229]. Moreover, secretion of oxalic acid was observed in the root apex of tomato plants which resulted in the formation of a Cd-oxalate complex and thus decreased the uptake of Cd [230]. A similar complex formation has also been observed between oxalate and Pb, which lowered the bioavailability of Pb in the roots of rice plants to provide Pb-stress tolerance [231]. In the Zn-hyperaccumulator plant *Thlaspi alpestre*, the accumulation of a higher concentration of Zn was correlated with the amount of malate [232,233]. Moreover, aerial parts of *A. helleri* showed sequestration of Zn via the formation of the Zn-malate complex [148]. Likewise, malate was also shown to be involved in the chelation and transfer of Cd into the vacuoles of *Solanum nigrum* leaves [234].

Amino acids, histidine, in particular, exhibit a strong affinity for metal ions such as Zn^2+^, Co^2+^, Ni^2+^, and Cu^2+^ and thus are involved in the direct chelation of these HMs. The capacity of *Alyssum* (Brassicaceae) plants to produce a high amount of histidine was reported to be directly correlated to the Ni-hyperaccumulation [235]. Moreover, the formation of the Zn-histidine complex has been observed in the roots of a Zn hyperaccumulator plant *Thlaspi caerulescens*, which allowed the plants to withstand a higher concentration of Zn [236]. Moreover, transgenic Arabidopsis plants with a 2-fold higher concentration of histidine showed improved Ni tolerance as compared with the wild type [237], further confirming a role of histidine in HM-stress tolerance.

In addition to these polyamines, organic acids, and amino acids, which are directly involved in the chelation of HMs, a rise in proline content has also been observed in many plant species including *Brassica juncea* [125], *Solanum melongena* [238], *Malva parviflora* [239], *Arachis hypogsea* [240], hybrid poplar (*Populus trichocarpa* × *deltoides*) [241], and *Groenlandia densa* under HM toxicity [242]. The proline thus produced is involved in providing stability and maintaining activities of various enzymes viz. nitrate reductase, protease, and ribonuclease as reported in rice [243]. Further, it was shown that exogenous applications of L-proline and betaine restored the membrane integrity and growth inhibition under Cd toxicity in cultured tobacco Bright Yellow (BY-2) cells [244]. Transgenic *Swingle citrumelo* plants carrying *pyrroline-5-carboxylate synthetase* (*P5CS112A)* genes showed higher production of endogenous L-proline which led to the enhanced expression of chloroplastic GR, cytosolic APX, and Cu/Zn SOD isoforms, indicating an intricate relationship between L-proline and enzymatic antioxidants [245]. Similarly, the role of glycinebetaine in tolerance to HM toxicity has also been highlighted. It was shown that foliar application of glycinebetain during Cr toxicity alleviated the toxic effects by reducing the Cr uptake and enhancing the antioxidants activity in different parts of mung bean enhances [246]. Moreover, the synthesis of osmoprotectants utilizes the excess reductants which provide NAD^+^ and NADP^+^ required for the regulation of respiration and photosynthesis, resulting in enhancement of biomass production, photosynthesis, promoted osmotic adjustment, and ROS scavenging systems in response to metals exposure [247].

Mannitol is a sugar alcohol that is capable of ROS scavenging activity, thus protecting the plants against oxidative stress induced by ·OH radicals [248]. [249] reported that Cd-tolerant lines of *Medicago truncatula* showed higher mobilization of total soluble sugars (glucose, fructose, and sucrose) than the Cd- susceptible lines, thus highlighting the role of mannitol in HM-stress tolerance. Similarly, the accumulation of mannitol along with other osmolytes including sucrose and glycinebetain has been observed in *Salvinia natans* plants exposed to a variety of HMs [250]. Further, exogenous application of mannitol was able to negate the toxic effects of Cr toxicity as reported in wheat [251] and maize [252], confirming a positive role of mannitol in HM-stress tolerance.

## 8. HM-Stress Tolerance Is Mediated by the Phytohormones Signaling

Phytohormones play a decisive role in stress tolerance of plants grown in metal-polluted via activation of a signaling cascade and antioxidant defense mechanism (Figure 3). For instance, Indole-3-butyric acid (IBA), IAA precursor, induced the antioxidant defense mechanism via NO signaling and decrease the ROS level by activating the GPX activity in barley roots as well as in tomato plants under Cd stress [253]. In *Oryza sativa*, the application of epibrassinolide (EBL) helps to plant combat Cr-induced oxidative stress following the upregulation of the antioxidant mechanism at the molecular level. Nevertheless, GR activity was found to be elevated when Cr-treated seedlings were exposed to EBL, resulting in the improvement of the growth of seedlings [254]. From the study, it was revealed that the Cr metal uptake and bioconcentration factor (BCF) content was significantly reduced under the influence of BR treatments. Hence, it is well known that phytohormones in metal tolerance such as brassinosteroids (BRs) play a key role in the mitigation of various stresses via inducing antioxidant defense system [255,256,257,258] and confer tolerance against HMs toxicity in various plants like rice, maize, tomato, wheat, Arabidopsis, radish, and mustard [259,260,261,262]. Their findings showed that treatment with EBL significantly decreased the amount of oxidative indicator MDA and H_2_O_2_ via elevating low molecular weight antioxidant ascorbic acid levels in Cr-treated seedlings [254]. Similarly, a recent study revealed that melatonin plays a protective role in melon root development in response to Cu toxicity by inhibiting the formation of jasmonic acid (JA). Pre-treatment of melatonin to melon seeds promoted excess Cu^2+^ chelation, reversed Cu-toxicity-induced root growth inhibition and the expression of genes related to the ROS-detoxification and cell wall modifications [263]. Moreover, the expression level of genes and metabolites were altered under melatonin treatment and involved in JA biosynthesis, suggesting a negative regulation of JA in HM-stress tolerance [137]. Further, the involvement of different phytohormones in HM-stress tolerance in plants has been excellently reviewed recently (Figure 3) [264].

## 9. Conclusions

HM toxicity is emerging as one of the most serious concerns across the globe. The toxic effects of HMs on plants depend on a variety of factors including type and concentration of HMs and plant species, plant growth stage, and exposure time. The negative effects of the HMs on the growth of the plants are the combined outcome of the changes in their anatomy and physiology. Tolerant plants retaliate to HM toxicity by activating a complex signaling network which often culminates into activation of a defense response that includes accumulation of (1) antioxidants for detoxification of excess ROS, (2) secondary metabolites for sequestration of HMs to vacuoles, and (3) compatible solutes for the osmoregulation along with regulation of metal transporters, among others. The information derived from the HM accumulator/hyperaccumulator plants can be used in the future to improve the ability of non-food plants to accumulate higher amounts of HMs in order to be used to mitigate the HM contamination in environmental matrices. Moreover, the information derived from the HM tolerant plants can be used to induce the mechanisms that result in reduced uptake and accumulation of HMs in food plants and thus their negative consequences for human health.

## Figures and Tables

**Figure 1 biomolecules-12-00043-f001:**
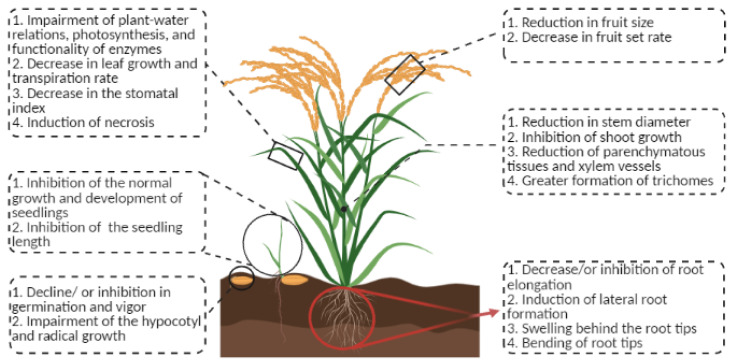
Image depicting the heavy metal (HM) toxicity-induced morphological, anatomical, and physiological changes in the plants.

**Figure 2 biomolecules-12-00043-f002:**
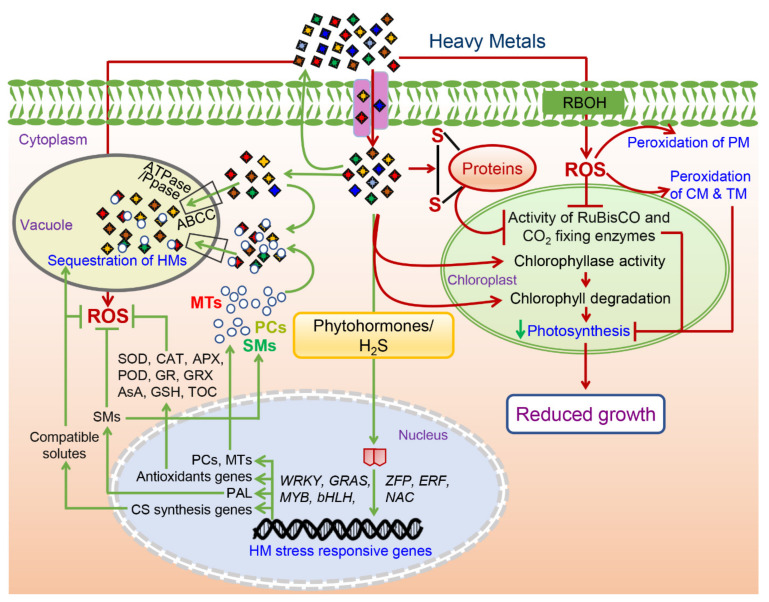
A putative diagram showing positive and positive molecular responses of the heavy metals (HM) toxicity in the plants. Responses marked with the red color represent negative effects of the HM toxicity while those marked with the green color represent tolerance response to alleviate the HM toxicity. Abbreviations: ROS—reactive oxygen species; SMs—secondary metabolites; CS—compatible solutes; PCs-phytochelatins; MTs—metallothioneins; SOD—superoxide dismutase; CAT—catalase; APX-ascorbate peroxidase; POD—peroxidase; GR—glutathione reductase; GRX—glutaredoxins; AsA—ascorbic acid; GSH—reduced glutathione; TOC—tocopherol; PAL—phenylalanine ammonia lyase; RBOH—respiratory burst oxidase homolog; PM—plasma membrane; CM—chloroplast membrane; TM—thylakoid membrane.

**Figure 3 biomolecules-12-00043-f003:**
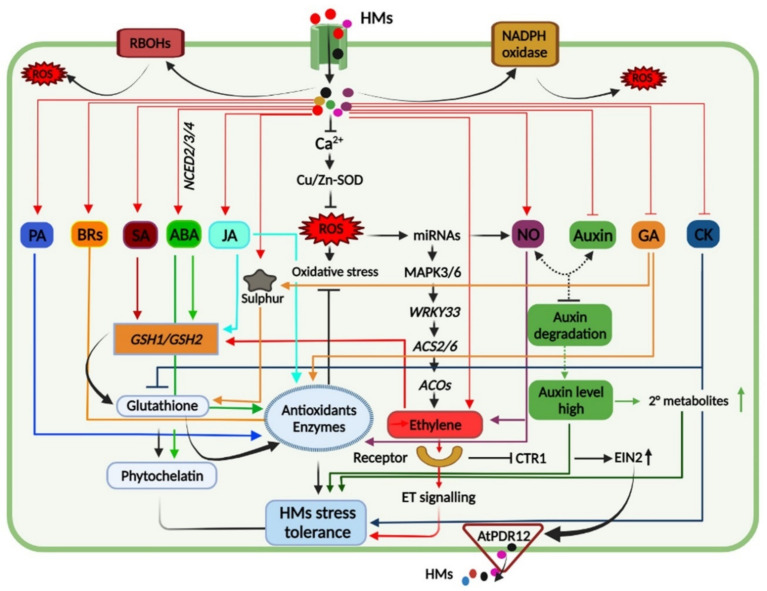
A graphical depiction highlighting interactions and crosstalk among phytohormones including abscisic acid (ABA), jasmonic acid (JA), salicylic acid (SA), brassinosteroids (BRs), polyamines (PA), ethylene, auxin, nitric oxide (NO), gibberellic acid (GA), and cytokinin (CK) under heavy metal exposure. HMs treatments increase endogenous levels of PA, BRs, SA, ABA, JA, ET, NO, and ROS amounts while the levels of auxin, GA, and CK were inhibited. The ROS produced in response to HM stress either by respiratory burst oxidase homolog (RBOH) activity and NADPH oxidase or by alteration in electron transport is also known to activate signal transduction. Alteration of these hormones and imbalance of ROS equilibrium leads to induction of antioxidant defense mechanism and HMs detoxification by promoting glutathione and phytochelatin biosynthesis, thus regulating HM stress tolerance to plants.

**Table 1 biomolecules-12-00043-t001:** List of the genes expressed under toxicities of different HMs.

Plant	Gene(s)	Metal(s)	Reported Phenotypes	References
*Lemna turonifera*	*AtNHX1*	Cadmium	vacuolar sequestration of metabolites and improved tolerance	Yao et al., 2020
*Triticum aestivum* L.	*TaCATs*	Arsenic	Stress tolerance	Tyagi et al., 2020
*Oryza sativa*	*CCoAOMT*	Copper	Lignin production and enhanced tolerance	Su et al., 2020
*Oryza sativa*	*cadA and bmtA*	Cadmium	Cd accumulation and Cd-nanoparticles (CdNPs) biosynthesis and improved tolerance by decreasing oxidative stress	Shi et al., 2020
*Hordeum vulagare*	*HvPAL, HvMDH and* *HvCSY*	Copper and Cobalt	Accumulation of phenolics and amino acids and increased tolerance	Lwalaba et al., 2020
*Jatropha curcas*	*JcMT2a and* *JcPAL*	Lead	Accumulation of antioxidants, e.g., flavonoids and phenolics and metal detoxification	Mohamed et al., 2020
Tobacco	*EhMT1*	Copper	Decreased hydrogen peroxide (H_2_O_2_) formation and increased tolerance	Xia et al., 2012
Tobacco	*TaMT3*	Cadmium	Increased superoxide dismutase (SOD) activity and conferred tolerance	Zhou et al., 2014
Tobacco	*OSMT1e-p*	Copper and Zinc	ROS scavenging and enhanced tolerance	Kumar et al., 2012
*Arabidopsis thaliana*	*BjMT2*	Copper and Cadmium	Inhibits root elongation but increased tolerance	Zhigang et al., 2006
*Hibiscus cannabinus* L.	*WRKY, GRAS, MYB, bHLH, ZFP, ERF, and NAC*	Cadmium	Enhanced tolerance via molecular mechanism	Chen et al., 2020
Tobacco	*NtCBP4*	Lead	Increased tolerance	Sunkar et al., 2000
*Arabidopsis thaliana*	*ACBP1*	Lead	Higher gene expression and enhanced tolerance	Xiao et al., 2008; Du et al., 2015
*Linum usitatissimum* L.	*LuACBP1 and LuACBP2*	Lead	Transcript level was higher in transgenic and improved tolerance	Pan et al., 2020
*Oryza sativa*	*OsSTAR1 and OsSTAR2*	Aluminium	Decreased aluminium level in cell wall and enhanced tolerance	Huang et al., 2020
*Fragaria vesca*	*FvABCC11*	Cadmium	Increased tolerance via ATP binding cassette (ABC) transporters	Shi et al., 2020
*Arabidopsis thaliana*	*AtABCC3 and AtABCC6*	Cadmium	Phytochelatin mediated tolerance during seedling development	Brunetti et al., 2015; Gaillard et al., 2008
*Oryza sativa*	*OsABCC1*	Arsenic	Increased tolerance via vacuolar sequestration	Song et al., 2014
*Arabidopsis thaliana*	*AtABCC1 and AtABCC2*	Cadmium and Mercury	Enhanced tolerance via vacuolar sequestration	Park et al., 2012
*Brassica napus*	*BnaABCC3 and BnaABCC4*	Cadmium	Enhanced stress tolerance	Zhang et al., 2018
*Triticum aestivum*	*TaABCC*	Cadmium	Distinct molecular expression and increased tolerance	Bhati et al., 2015

## Data Availability

Not applicable.

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
