# Peer review of "A Comprehensive Review on the Heavy Metal Toxicity and Sequestration in Plants"

_biomolecules, 2021, doi:10.3390/biom12010043_

Round 1
Reviewer 1 Report
The Ms by Riyazuddin et al. deals with an overview on the most recent findings in the topic of both the toxic effects exerted by heavy metals in plants and the plant defense responses associated to the increased presence of HMs in the environment. Even if a large body of literature is present on the matter, also reviewing the state of the art, the topic is so relevant that a focus on the mechanism of actions, processes and defence responses induced in plants by HMs is to be appreciated. In this regard, the present Ms addresses in a clear and effective way the most relevant physiological processes affected by HMs in plants and the main counteractive actions exerted by plants to cope with the increase levels of such ions in their cells. Therefore, in my opinion, the Ms deserves to be published after a revision step addressing some remarks listed below.
L42 The point related to the toxicity level for chemical elements not playing in plants any physiological function like Hg and Cd is controversial and in this regard the statement about the “high concentration” should be better argued
L53 The topic about soils contaminated by HM and their possible accumulation in food plants with the consequences for human health deserves two or three more lines
L73 The statement about bioindicator plants for HMs and the damaging effects on plants sounds misleading as bioindicator plants in the concept expressed by Baker (1981) “are seen as a further mode of response where proportional relationships exist between metal levels in the soil, uptake and accumulation in plant parts”, so without reference to toxicity effects. I suggest to clarify it
L170 Change “alters” with “alter”
L176 Not clear if “light” means “reduce”. If so, better use “reduce”
L177 Stomatal conductance and water uptake are not considered in Figure 2. Please check and amend
L236 I think that a fourth way is to be taken into account, i.e. a direct toxic action of HMs on antioxidant enzymes. See literature on that
L292 Modify the formula H2S writing the number as subscript
L304-319 Check and correct the character size
L 305 and 307 Spell out MT and PC at first appearance
L339 Not essential nutrients, such as the most toxic HMs, are commonly not transported in not hyperaccumulating plants and the highest concentration is usually found in the roots. Better clarify this sentence
L342-352 I think this paragraph should be revised to better describe the different physiological processes found in tolerant/hypertolerant plants and accumulator/hyperaccumulator plants
L354 The statement “another strategy” is misleading as no strategy to limit the absorption of HMs is described in the above reported paragraph
L475 and following. Some of these compounds, as polyamines, aminoacids (like histidine) and organic acids, are claimed to be actively involved in HM detoxification as chelating agents, antioxidative response inducers, stress signaling molecules besides osmoregulation (as for paragraph heading). Therefore, I suggest to mention them within the discussion made in the previous paragraph. In this context, I would like to underline the role of silicon to alleviate the HM toxicity in plants by different mechanisms involving restriction of metal uptake and transport, improving antioxidative response and HM compartmentation. Authors could profit from the large body of literature dealing with this matter to add a brief paragraph on it.
L546-554 The conclusion section should be revised to be improved with future perspective in the field, as for example about the reduction of HM accumulation in food plants and the consequences for human health and the improvement of the ability for non food plants to accumulate metals in order being used to mitigate the HM contamination in environmental matrices. Moreover, considering the discussion along the Ms, the role of the phytohormone signaling seems too much emphasized.
Author Response
The Ms by Riyazuddin et al. deals with an overview on the most recent findings in the topic of both the toxic effects exerted by heavy metals in plants and the plant defense responses associated to the increased presence of HMs in the environment. Even if a large body of literature is present on the matter, also reviewing the state of the art, the topic is so relevant that a focus on the mechanism of actions, processes and defence responses induced in plants by HMs is to be appreciated. In this regard, the present Ms addresses in a clear and effective way the most relevant physiological processes affected by HMs in plants and the main counteractive actions exerted by plants to cope with the increase levels of such ions in their cells. Therefore, in my opinion, the Ms deserves to be published after a revision step addressing some remarks listed below.
Reply: Thank you for your positive feedback on the manuscript and for your valuable comments. We have now modified the manuscript as per your comments.
L42 The point related to the toxicity level for chemical elements not playing in plants any physiological function like Hg and Cd is controversial and in this regard the statement about the “high concentration” should be better argued
Reply: Thank you for your suggestion. We have now modified the sentence to avoid any confusion.
L53 The topic about soils contaminated by HM and their possible accumulation in food plants with the consequences for human health deserves two or three more lines
Reply: Thank you for your suggestion. We have now included the required information in the revised manuscript (Lines 59-63).
L73 The statement about bioindicator plants for HMs and the damaging effects on plants sounds misleading as bioindicator plants in the concept expressed by Baker (1981) “are seen as a further mode of response where proportional relationships exist between metal levels in the soil, uptake and accumulation in plant parts”, so without reference to toxicity effects. I suggest to clarify it
Reply: Thank you for pointing out the incorrect statement. We have now modified this sentence for better clarity (Lines 84-85)
L170 Change “alters” with “alter”
Reply: Corrected.
L176 Not clear if “light” means “reduce”. If so, better use “reduce”
Reply: This sentence is now modified as “Literature hints that the impact of HMs on photosynthetic machinery depends on the reactivity and concentration of HMs in leaves which subsequently affects light capture, electron transport, stomatal conductance, and water uptake”.
L177 Stomatal conductance and water uptake are not considered in Figure 2. Please check and amend
Reply: We are sorry for adding the incorrect statement. We have now modified the statement to reflect the content of the figure 2.
L236 I think that a fourth way is to be taken into account, i.e. a direct toxic action of HMs on antioxidant enzymes. See literature on that (mechanism? How they interfere with antioxidant enzymes?)
Reply: Thank you for the suggestion. We have now included two more ways in to the list including (4) directly inhibiting the activities antioxidant enzymes by exchanging essential cations from the specific binding sites and targeting their -SH groups, and (5) by inducing the activity of NADPH oxidases (lines 260-262).
L292 Modify the formula H2S writing the number as subscript
Reply: Corrected.
L304-319 Check and correct the character size
Reply: Thank you for pointing out the error. We have now corrected it in the revised manuscript.
L 305 and 307 Spell out MT and PC at first appearance
Reply: We have introduced the abbreviations of both PC and MT in line number 69.
L339 Not essential nutrients, such as the most toxic HMs, are commonly not transported in not hyperaccumulating plants and the highest concentration is usually found in the roots. Better clarify this sentence
The major percentage of the HMs in plants is transported from root to stem and thus the concentration of HMs in the aboveground parts is higher than the roots [77].
Reply: Thank you for pointing out the error. We have now revised this section as follows (Lines 368-372):
In most of the cases, the major percentage of the HMs in plants is transported from root to stem and thus the concentration of HMs in the aboveground parts is higher than the roots, however, the most toxic HMs, are commonly not transported in non-hyperaccumulating plants and the highest concentration is usually found in the root. For instance, in tomato plants, the highest concentrations for Cu, Ni, Cr, Mn and Pb were reported in the following order of root>leaf>stem>fruit (Arslan et al., 2020).
L342-352 I think this paragraph should be revised to better describe the different physiological processes found in tolerant/hypertolerant plants and accumulator/hyperaccumulator plants
HM stress-tolerant plants get rid of the unused and extra amount of metal ions by effluxing and/or compartmentalization majorly in the vacuole with the help of two vacuolar proton pumps including an ATPase and a Ppas (Figure 2) [140–142]. Among various other HMs, Zn is majorly sequestrated in vacuoles of epidermal cells and trichomes of mesophyll cells as shown in Thlaspi caerulescens [143] and in Arabidopsis helleri respectively [144,145]. In addition, Zn is also found to be accumulated, although to a lesser extent in the cell wall and cytosol in leaves of P. griffithii [146]. Similar findings were also reported in the case of Ni where vacuole was found to be a major Ni accumulating region in a Ni hyperaccumulator plant Alyssum serpytllifolium [147]. In the case of Cd, epidermal and mesophyll cells have been identified as major Cd accumulating cells as reported in B. juncea and Silene vulgaris [148–150], and in A. halleri and Brassica napus, respectively [145,151].
Reply: We have modified this paragraph as per the suggestion (lines 373-388).
L354 The statement “another strategy” is misleading as no strategy to limit the absorption of HMs is described in the above reported paragraph
Reply: Corrected
L475 and following. Some of these compounds, as polyamines, aminoacids (like histidine) and organic acids (citric acid, melvic acid etc…), are claimed to be actively involved in HM detoxification as chelating agents, antioxidative response inducers, stress signaling molecules besides osmoregulation (as for paragraph heading). Therefore, I suggest mentioning them within the discussion made in the previous paragraph. In this context, I would like to underline the role of silicon to alleviate the HM toxicity in plants by different mechanisms involving restriction of metal uptake and transport, improving antioxidative response and HM compartmentation. Authors could profit from the large body of literature dealing with this matter to add a brief paragraph on it.
Reply: Thank you for your suggestion. We have now included the suggested details in the revised manuscript (lines 409-417 and 528-580)
L546-554 The conclusion section should be revised to be improved with future perspective in the field, as for example about the reduction of HM accumulation in food plants and the consequences for human health and the improvement of the ability for non food plants to accumulate metals in order being used to mitigate the HM contamination in environmental matrices. Moreover, considering the discussion along the Ms, the role of the phytohormone signaling seems too much emphasized.
Reply: Thank you. We have now revised the conclusion section following your suggestions.
Reviewer 2 Report
- The introduction seems bit week in explaining the different metals toxicity and its effects. Some of the important below manuscript need to be cited to give more information about the mechanism of different metals toxicity and its significance in plants.
https://link.springer.com/article/10.1007/s00449-015-1447-y
https://www.frontiersin.org/articles/10.3389/fenvs.2021.604216/full
https://academic.oup.com/jimb/article/45/1/31/5996653
https://link.springer.com/article/10.1007/s12257-020-0218-z
- Please include a table comprising a potential list of heavy metal toxicity and sequestration techniques available. It will be useful for the readers.
Author Response
The introduction seems bit week in explaining the different metals toxicity and its effects. Some of the important below manuscript need to be cited to give more information about the mechanism of different metals toxicity and its significance in plants.
Kannan Maruthamuthu, M., Ganesh, I., Ravikumar, S., & Hong, S.H. (2015a). Evaluation of zraP gene expression characteristics and construction of a lead (Pb) sensing and removal system in a recombinant Escherichia coli. Biotechnology Letters, 37(3), 659-664.
https://link.springer.com/article/10.1007/s00449-015-1447-y
kannan Maruthamuthu, M., Nadarajan, S.P., Ganesh, I., Ravikumar, S., Yun, H., Yoo, I.K., & Hong, S.H. (2015b). Construction of a high efficiency copper adsorption bacterial system via peptide display and its application on copper dye polluted wastewater. Bioprocess and biosystems engineering, 38(11), 2077-2084.
https://www.frontiersin.org/articles/10.3389/fenvs.2021.604216/full
González-Henao, S., & Ghneim-Herrera, T. (2021). Heavy metals in soils and the remediation potential of bacteria associated with the plant microbiome: evidence points to Klebsiella and Enterobacter as the best candidates for bioremediation and bacteria-assisted phytoremediation strategies in soils contaminated with arsenic, cadmium, and lead. Frontiers in Environmental Science, 9of 15.
https://academic.oup.com/jimb/article/45/1/31/5996653
Maruthamuthu, M.K., Selvamani, V., Nadarajan, S.P., Yun, H., Oh, Y.K., Eom, G.T., & Hong, S.H. (2018). Manganese and cobalt recovery by surface display of metal binding peptide on various loops of OmpC in Escherichia coli. Journal of Industrial Microbiology and Biotechnology, 45(1), 31-41.
https://link.springer.com/article/10.1007/s12257-020-0218-z
Wang, Y., Selvamani, V., Yoo, I.K., Kim, T.W., & Hong, S.H. (2021). A Novel Strategy for the Microbial Removal of Heavy Metals: Cell-surface Display of Peptides. Biotechnology and Bioprocess Engineering, 1-9.
Reply: Thank you for your suggestion. We looked into the suggested references, however none of these are related to the plants, yet we have tried to cite these at the relevant sections.
Please include a table comprising a potential list of heavy metal toxicity and sequestration techniques available. It will be useful for the readers.
Reply: Thank you for your suggestion, however, in this manuscript, we are only focusing on the sequestration of the HMs by the plants and thus the details and discussion on different techniques available for the HM sequestration are out of the scope of this manuscript.
Reviewer 3 Report
The title can be improved. “A mechanical insight” looks like a representation of a single hypothesis instead of multiple mechanisms discussed in the article. The present title is also ok but still, authors can try to improve so that it will reflect the correct sense of the MS.
Abstract
“The accretion of heavy metals (HMs) because of the increased anthropogenic activities has 15 caused a severe threat to the environment and human health” – not clear, need improvement.
Abstract also missing the take-home message. Authors have introduced the Heavy metal toxicity and mechanisms known in plants and at the end of the abstract stated – “This comprehensive review provides mechanistic insights towards a wider aspect of HM toxicity, ranging from the negative impact of HMs on plant growth and regulation to the mechanisms employed by the plants to alleviate the HM toxicity” - But what new information provided in the article is not highlighted in the abstract.
Introduction
In the first paragraph, better to enlist heavy metals that are really a concern for crop production. Authors have listed Fe and Zn but their toxicity is hardly a concern.
Figure 1 – several issues with figure 1, this figure does not bring any information. Heavy metals affect the overall growth of the plant and include everything from fruit/grain developments, root growth, plant height. Therefore this figure is useless here. Secondly, the abbreviation HM needs to be introduced, figure legend is always standalone. Similarly, the same information is provided in table 1.
The paragraph starting with “Roots are in direct contact…” needs to revise. The provided information is not in the flow and several unrelated aspects have been discussed together. For root growth and hormonal regulation, keep two separate paragraphs.
Figure 2 – This figure is informative but needs to provide specific information for major heavy metals. The uptake and sequestration for each HM is specific. I suggest denoting different HMs with different symbols/legends.
Abbreviations used in the figure need to be explained in the figure legend.
I suggest not using abbreviations in the subheadings
Authors can add a section describing the role of beneficial elements like silicon in alleviating heavy metal toxicity in plants.
Author Response
The title can be improved. “A mechanical insight” looks like a representation of a single hypothesis instead of multiple mechanisms discussed in the article. The present title is also ok but still, authors can try to improve so that it will reflect the correct sense of the MS.
Reply: Thank you for the suggestion. We have now modified the title as: A Comprehensive Review on the Heavy Metal Toxicity and Sequestration in Plants. We hope the modified title would better reflect the contents of this manuscript.
Abstract
“The accretion of heavy metals (HMs) because of the increased anthropogenic activities has 15 caused a severe threat to the environment and human health” – not clear, need improvement.
Reply: Thank you. We have now modified this sentence to make it clearer.
Abstract also missing the take-home message. Authors have introduced the Heavy metal toxicity and mechanisms known in plants and at the end of the abstract stated – “This comprehensive review provides mechanistic insights towards a wider aspect of HM toxicity, ranging from the negative impact of HMs on plant growth and regulation to the mechanisms employed by the plants to alleviate the HM toxicity” - But what new information provided in the article is not highlighted in the abstract.
Reply: We have now modified the abstract as per the suggestion (Lines 30-33). Thank you
Introduction
In the first paragraph, better to enlist heavy metals with references that are really a concern for crop production. Authors have listed Fe and Zn but their toxicity is hardly a concern.
Reply: Thank you for the suggestion. We have now removed Fe and Zn from the list and modified the subsequent statement to avoid any confusion (Lines 43-48).
Figure 1 – several issues with figure 1, this figure does not bring any information. Heavy metals affect the overall growth of the plant and include everything from fruit/grain developments, root growth, plant height. Therefore, this figure is useless here. Secondly, the abbreviation HM needs to be introduced, figure legend is always standalone. Similarly, the same information is provided in table 1.
Reply: Thank you for the suggestion. We have removed Table 1 from the manuscript to avoid any possible duplication of the content between figure 1 and table 1. However, we request to please allow us to retain figure 1 as this figure provides a quick overview of the HM toxicity induced changes in plants. In addition, we have also provided the abbreviation of the HM in the figure legend and tried to improvise the overall legend to merge with the manuscript content as per the suggestion.
The paragraph starting with “Roots are in direct contact…” needs to revise. The provided information is not in the flow and several unrelated aspects have been discussed together. For root growth and hormonal regulation, keep two separate paragraphs.
Reply: Thank you for the suggestion. We have now revised this paragraph and separated the root growth and hormonal regulation paragraph in two paragraphs.
Figure 2 – This figure is informative but needs to provide specific information for major heavy metals. The uptake and sequestration for each HM is specific. I suggest denoting different HMs with different symbols/legends.
Reply: Thank you for the suggestion. We have tried to make it metal specific, however, the final image is too crowded and is difficult to interpret. Therefore, we request to please allow us to use the same figure showing an overview of the changes taking place under HM toxicity.
Abbreviations used in the figure need to be explained in the figure legend.
I suggest not using abbreviations in the subheadings
Reply: Thank you for pointing out the missing information. We have now explained all the abbreviations used in the figure legend as per the suggestion.
Authors can add a section describing the role of beneficial elements like silicon in alleviating heavy metal toxicity in plants.
Reply: Thank you for your suggestion. We have now added the role of silicon and selenium in alleviating HM toxicity in plants as per the suggestion (Lines 409-417).
Round 2
Reviewer 1 Report
The revised version of the Ms by Riyazuddin et al. satisfactorily addressed the remarks made to the original submission. Therefore, it meets the requests for publication. Only two editing corrections should be made regarding: 1) L191, correct the singular form; 2) L321, the number in the formula should be written as subscript not reducing the character size.
Reviewer 3 Report
The revised version of the MS looks fine.